# Effects of 4-Week Creatine Supplementation Combined with Complex Training on Muscle Damage and Sport Performance

**DOI:** 10.3390/nu10111640

**Published:** 2018-11-02

**Authors:** Chia-Chi Wang, Chu-Chun Fang, Ying-Hsian Lee, Ming-Ta Yang, Kuei-Hui Chan

**Affiliations:** 1Physical Education Office, National Taipei University of Business, Taipei 10051, Taiwan; sunnywang@ntub.edu.tw (C.-C.W.); kammyfang@ntub.edu.tw (C.-C.F.); 2Graduate Institute of Athletics and Coaching Science, National Taiwan Sport University, Taoyuan 33301, Taiwan; 1040303@ntsu.edu.tw; 3Centre for General Education, Taipei Medical University, Taipei 10031, Taiwan; yangrugby@gmail.com

**Keywords:** creatine kinase, optimal individual post-activation potentiation time, half squat, plyometric jump

## Abstract

Creatine supplementation has an ergogenic effect in an acute complex training bout, but the benefits of chronic creatine supplementation during long-term complex training remain unknown. The study aimed to evaluate the effects of 4-week complex training combined with creatine supplementation on sport performances and muscle damage biomarkers. Thirty explosive athletes were assigned to the creatine or placebo group, which consumed 20 g of creatine or carboxymethyl cellulose, respectively, per day for 6 days followed by 2 g of the supplements until the end of the study. After 6 days of supplementation, subjects performed tests of one repetition maximum (1-RM) strength of half squat and complex training bouts to determine the optimal individual post-activation potentiation time. Thereafter, all subjects performed a complex training programme consisting of six sets of 5-RM half squats and plyometric jumps 3 times per week for 4 weeks. Body composition, 30-m sprint and jump performances were assessed before and after the training period. Moreover, blood creatine kinase (CK) activity was analysed at the first and the last training bout. After the training, the 1-RM strength in the creatine group was significantly greater than in the placebo group (*p* < 0.05). CK activity after the complex training bout in the creatine group was significantly reduced compared with the placebo group (*p* < 0.05). No differences were noted for other variables. This study concluded that creatine supplementation combined with complex training improved maximal muscular strength and reduced muscle damage during training.

## 1. Introduction

In recent years, many athletes have used ergogenic aids to maintain body conditioning, enhancing recovery and physiological adaptations during long-term training programmes. Therefore, the efficacy of ergogenic aids has always attracted great attention, and numerous researchers have sought to combine ergogenic aid and exercise training programmes to reinforce the benefits of training. Creatine (Cr) is a popular ergogenic aid among athletes at all levels. Many studies demonstrate that chronic Cr supplementation in combination with resistance training is more effective at augmenting training workouts and increasing muscular strength and lean body mass [1,2].

Complex training is now recognized as an effective method for integrating strength and explosive power development [3]. Complex training generally involves the execution of heavy-resistance exercise (HRE) (1–5 repetition maximum (RM) strength) followed by a brief rest interval. Then, biomechanically similar explosive exercises are executed. Post-activation potentiation (PAP) has been evidenced as the physiological rationale for complex training. HRE will induce PAP, which is the change in force-time and force-velocity characteristics of skeletal muscle, leading to increased power output and, consequently, increasing the performance of the explosive exercise [4,5,6,7]. The effects of complex training have been supported by most scientifically controlled studies evaluating its chronic development effects on maximum strength, explosive power, sprint and sports performance compared with other training methods [4,8,9]. However, some studies present inconsistent results. For example, the study of Mihalik et al. [10] revealed no significant difference on vertical jump height and power output between 4 weeks of complex training and compound training. MacDonld et al. [11] revealed a significant increase in muscle strength by resistance training, plyometric training and complex training, but no differences were noted between the training modes. Several systematic reviews summarize the primary factors that affect the consequences of PAP, including subject characteristics, preconditioning exercise, fatigue recovery and individualized phenomenon [7,12].

Our previously studies show that PAP is a highly individualized phenomenon in high school and university athletes [13,14]. Our previously studies also demonstrated that a 6-day loading phase of Cr supplementation enhances upper and lower body maximal strength, benefits fatigue recovery and reduces optimal individual PAP time for training efficiency during a complex training bout [13,14]. The mechanism by which Cr supplementation produces better effects on chronic adaptation to a resistance training programme may be due to maximizing the potential and advantages of the phosphocreatine energy system in skeletal muscle, thereby allowing athletes to sustain high-intensity training [15]. Consequently, it may be hypothesized that Cr supplementation may help to enhance athletes’ abilities more effectively during several weeks of a complex training programme.

In another aspect, resistance training elicits a milieu of acute physiological responses and chronic adaptations that are critical for increasing muscular strength, power, hypertrophy and local muscular endurance [16]. Therefore, chronic complex training with Cr supplementation might promote more positive effects on muscle mass, muscular strength and power development. In addition, complex training is a repeated bout of high-intensity and eccentric resistance exercise. It is well known that eccentric resistance exercise induces muscle damage. Serum creatine kinase (CK) has been generally considered to be an indirect biomarker of muscle damage due to its ease of identification and the relatively low cost of assays to quantify it. In spite of CK rising after exercise being highly variable and affected by individual factors and exercise variable, the evaluation of CK pre- and post-exercise may provide a diagnostic tool for the detection of muscle damage, with much less invasiveness than required in a muscle biopsy [17]. Recently, some findings indicated that Cr supplementation might reduce post-exercise muscle damage via mechanisms stabilizing the sarcolemma [18] and regulating mitochondrial permeability [19]. Studies demonstrated the positive effects of Cr supplementation on reducing CK after resistance training [20,21]. However, Cr supplementation could not reduce muscle damage from high-repetition exercise [22,23,24]. The property of complex training is close to resistance training, and its repetitions are lower than those previous studies. Therefore, Cr supplementation may benefit the muscle damage induced by complex training.

The combined effects of Cr supplementation and chronic complex training have not been reported in previous studies. Therefore, this study investigated the effect of long-term Cr supplementation on biomarkers of muscle damage, muscle strength, sports performance and body composition after 4 weeks of complex training. We hypothesized that long-term Cr supplementation would increase sports performance, reduce muscle damage and exhibit a positive effect on body composition after 4 weeks of complex training.

## 2. Experimental Section

### 2.1. Research Design

This study is the second stage of our previous study investigating the combined effect of 4-week Cr supplementation and complex training. Therefore, a double-blind, randomized matched-pair design was used to assign 30 subjects from 3 explosive-type sports into a Cr or placebo (Pla) group based on the procedures described in the previous study [14]. Subjects consumed 20 g of creatine or carboxymethyl cellulose, respectively, per day for 6 days followed by 2 g of the supplements until the end of the study. After a 6-day loading phase, all subjects executed 1-RM half squat for resistance loading and optimal individual PAP time tests on different days. Then, the body composition, 30-m sprint and jump performance were also determined. After the pre-tests were completed, all subjects started the complex training with optimal individual PAP times 3 times per week for 4 weeks. The blood samples were collected to analyse the activities of CK at the first and the last training bouts. Two days after the last training, the body composition, 1-RM half squat, 30-m sprint and jump performance were determined again. The Institutional Review Board of the Taoyuan General Hospital, Taiwan approved this study.

### 2.2. Subjects

Thirty male university athletes from baseball, basketball, and tchoukball teams (10 for each sport) volunteered to participate in this study. The sample size computation was based on the study by Flanagan and Jakeman [25]. Based on a statistical power analysis, a total sample size of 16 participants (8 per group) was needed to achieve a statistical power of 0.8 to detect a large effect size (ES) for supplement-time interaction at an alpha level of 0.05 [26]. The characteristics of the subjects are described in Table 1. No significant differences were noted for any variable. All subjects provided written informed consent before participation. This study was conducted in the off-season period and all subjects maintained their basic training programmes with sport-specific skills. Subjects were asked to maintain their normal diet pattern during the experimental period. Subjects were excluded if they had one of the following: (1) a maximal squat strength less than 1.5-fold their body weight [27]; (2) injury to a lower limb within the past 6 months; (3) experience with both half squat and plyometric training within less than 1 year; or (4) consumed chronic or daily doses of anti-inflammatory medications or nutritional supplements within the past month.

### 2.3. Complex Training Protocol

The complex training consisted of six sets of HRE and plyometric training, and the rest time between sets was 4 min. All subjects performed a 5-RM of half squat, rested for optimal individual PAP time, and then conducted vertical jump or squat jump for 8 repetitions. The complex training protocol is presented in Table 2.

### 2.4. Supplement Protocol

After grouping, subjects in the Cr group began consuming 5 g of Cr monohydrate (creatine fuel powder; Twinlab Corp, Hauppauge, NY, USA) plus 5 g of dextrose dissolved in 300 mL of water 4 times (after breakfast, lunch, and dinner as well as before bedtime) per day for 6 days. Subjects in the Pla group followed the same dosage and protocol but consumed carboxymethyl cellulose (food grade powder, Sheng Yuang Food Industrial Co., Tucheng, New Taipei City, Taiwan) instead of Cr. The supplements for both groups were the same colour and taste. For maintenance, subjects ingested single daily doses of 2 g of creatine monohydrate or carboxymethyl cellulose powder plus 2 g dextrose after lunch during the 4-week complex training programme and until the post-tests of the study.

### 2.5. Optimal Individual Post-Activation Potentiation (PAP) Time and Jump Performance Tests

The countermovement jumps (CMJ) test by SmartJump (Fusion Sport, Brisbane, Australia) is an acceptable method for jump performance measuring [28]. After performing low-intensity aerobic exercise at a comfortable pace followed by a lower limb light stretching exercise for warm-up, subjects performed the CMJ test with a 5-second rest interval, and the best attempt of the two jumps was recorded as baseline. During the CMJ test, subjects jumped on a contact mat (SmartJump, Fusion Sport, Brisbane, Australia) with hands placed on their hips at all times, and a downward counter movement until the knee angle was approximately 90° was performed. After a 5-min rest, subjects executed a set of complex training bouts with 5-RM half squat exercises to elicit PAP [29] followed by CMJs with a counterbalanced order of the 6 rest intervals (1, 3, 5, 7, 9, 11 min or 2, 4, 6, 8, 10, 12 min) for 2 days. The optimal individual PAP time was the rest interval with the maximum delta-values of jump height for complex training bouts minus baseline values.

### 2.6. Sprint Tests

A Newtest Powertimer (Oulu, Finland) was applied to the sprint tests. The testing system was shown to be a useful instrument for running speed [30]. The participants executed two maximal 30-m sprints with 10-min recovery periods in between. The evaluation started 5 min after the specific warm-up. Running time was recorded using two photocell gates (Newtest Powertimer, Oulu, Finland) positioned at the starting line and 30 m. The participants commenced the sprint when ready behind the starting line. The fastest 30-m sprint time was selected for analysis. 

### 2.7. One Repetition Maximum (1-RM) Estimation

Prediction of 1-RM strength for the half squat was determined based on the protocol described by Baechle et al. [31]. In brief, subjects jogged for 5 min on a treadmill followed by lower/upper limb light stretching exercises and two light resistance warm-up sets. After 1 min of rest, the subjects executed a load of 87%~93% of the predicted 1-RM through the full range of motion. After each successful performance, the load was increased in increments of 14–18 kg until only one successful repetition could be completed. Four-min rests were given between each test. The increase or decrease in the load continued until the subject was able to complete one repetition with the proper exercise technique. Ideally, the subject’s 1-RM was measured within five testing sets.

### 2.8. Anthropometric Measurements

All subjects visited the laboratory in the morning for anthropometric measurements included body height, body mass, fat-free mass and body fat percentage. Standing body height without shoes and socks was measured to the nearest 0.1 cm with a height scale. Body mass, lean body mass and body fat percentages of participants were measured using a bioelectrical impedance instrument (InBody 3.0, Biospace, Seoul, Korea) with standard methods to assess body composition.

### 2.9. Blood Creatine Kinase (CK) Analysis

Before the experiment for complex training bouts, the subjects rested for approximately 10 min in a seated position, and approximately 50-μL capillary blood from the fingertip of the subjects was collected into a heparinized capillary tube using an automatic lancet device. After the complex training bout, blood samples from the fingertip were also drawn immediately, 24-h, and 48-h post-exercise. Blood samples were centrifuged at 3000 rpm for 3 min at room temperature and then immediately pipetted to CK test slide (Fuji Dri-Chem Slide CPK, Fujifilm, Tokyo, Japan) and analysed using the Fuji Dri-Chem analyser system (Fuji Dri-Chem 4000i, Fujifilm, Tokyo, Japan).

### 2.10. Statistical Analysis

Statistical analyses were performed using SPSS version 21.0 software (SPSS Inc., Chicago, IL, USA). Data are expressed as the mean ± standard deviation (SD) and 95% confidence interval (95% CI). An independent-sample *t*-test was used to compare the subjects’ characteristics between the groups. A mixed design two-way analysis of variance (ANOVA) (group × time) was used to compare the variables of body mass, fat-free mass, body fat percentage, 1-RM strength, performances of 30-m sprint, jump height and peak power between two groups before and after the 4-week of training programme. A mixed design three-way ANOVA (group × training bout × time point) was used to compare the differences in CK of two groups at different time points based on the first and the last complex training bouts. Cohen’s conventions for ES (Cohen d) were calculated by the G*Power 3.1 software program (Heinrich-Heine-Universität, Düsseldorf, Germany), where the ESs of 0.2, 0.5, and 0.8 are considered small, medium, and large, respectively. Statistical significance was set as *p* < 0.05.

## 3. Results

### 3.1. Body Composition and Sport Performances

Table 3 presents the body composition and sport performance results for the Cr or Pla groups before and after 4 weeks of training. After training, the 1-RM strength in both the Cr and Pla groups significantly increased compared with pre-training results (178.33 ± 16.86 vs. 133.67 ± 14.07 kg for the Cr group (*p* < 0.05, ES = 2.85) and 165.66 ± 14.62 vs. 131.67 ± 15.77 kg for the Pla group (*p* < 0.05, ES = 2.23)). The 30-m sprint performance in both the Cr and Pla groups significantly improved after training compared with pre-training (4.18 ± 0.18 vs. 4.33 ± 0.20 seconds for the Cr group (*p* < 0.05, ES = 0.42) and 4.19 ± 0.16 vs. 4.27 ± 0.21 seconds for the Pla group (*p* < 0.05, ES = 2.85)). Moreover, the strength in the Cr group was significantly greater compared with the Pla group after training (178.33 ± 16.86 kg vs. 165.66 ± 14.62 kg, *p* < 0.05, ES = 0.80). No significant differences in other variables of body composition and sport performance were noted between pre- and post-training in both groups and between groups (*p* > 0.05). However, after 4 weeks of training, the body fat percentage, jump height, and jump peak power for all subjects were significantly improved compared with that before training (*p* < 0.05).

### 3.2. Muscle Damage

Muscle damage responses (CK activities) in the Cr and placebo groups after the first and the last training bouts are presented in Figure 1. The CK activities immediately after and 24 and 48 h after the training were significantly increased in both Cr and Pla groups, as well as at the first and the last training bouts. However, at both the first and the last training bouts, the CK activities after 24 and 48 of the complex training in the creatine group were significantly reduced compared with the Pla group (285.40 ± 77.58 vs. 347.13 ± 93.57 U/L, respectively, for 24 h after training at the first training bout (*p* < 0.05, ES = 0.72); 262.40 ± 68.55 vs. 329.80 ± 76.13 U/L for 24 h after training at the last training bout (*p* < 0.05, ES = 0.93); 215.33 ± 65.55 vs. 226.26 ± 88.76 U/L for 48 h after training at the first training bout (*p* < 0.05, ES = 0.12); and 167.13 ± 59.13 vs. 229.80 ± 58.15 U/L for 48 h after training at the last training bout (*p* < 0.05, ES = 0.83)).

## 4. Discussion

This is the first study to evaluate the effects of Cr supplementation during 4 weeks of complex training with optimal individual PAP time on sport performance, body composition, and muscle damage. The major findings of the present study were that the 4 weeks of complex training with optimal individual PAP time (3 times per week) during the off-season period could reduce body fat percentage and enhance maximal muscular strength, performance of 30-m sprint, jump height and peak power. Moreover, creatine supplementation during the complex training protocol applied in this study could increase maximal muscular strength after 4 weeks of training and reduce the muscle damage caused by the complex training bout. 

This study applied optimal individual PAP time in the complex training programme based on our observation that PAP is a highly individualized phenomenon [13,14]. The results of this study demonstrated that after 4 weeks of complex training during the off-season period, subjects significantly increased their 1-RM strength of half squat (Cr group: +33.4%; Pla group: +25.8%), and creatine supplementation combined with complex training was more effective for maximum strength development. Studies suggest that a complex training [9,32,33] or resistance training programme combined with Cr supplementation [34,35,36] can enhance the 1-RM performance. Our result is consistent with the results of Vandenberghe et al. [36], indicating that the 1-RM squat of the Cr and placebo groups increased 45.6% and 24.5%, respectively, after 10 weeks of resistance training. Moreover, the study by Arciero et al. [37] demonstrated that leg press strength increased 42% after 4 weeks of resistance training with Cr supplementation. A systematic review and meta-analyses concluded that Cr supplementation is an effective method in lower-limb strength performance for exercise with a duration of less than 3 min [38]. The increased muscle strength following Cr ingestion plus resistance training could result from several mechanisms, including alteration in the expression of myogenic transcription factors, an increase in satellite cell mitotic activity, or simply an increase in the intensity of individual workouts resulting from a better match between ATP supply and demand during exercise [39].

Our study indicates that 4 weeks of complex training and creatine supplementation does not affect body mass and fat-free mass, but 4 weeks of complex training reduces body fat percentage. This observation is similar to the findings of Chilibeck et al. [40], indicating that Cr supplementation had no effect on body mass, body fat and lean tissue mass during season training. This result does not support our hypothesis, which is based on the positive effect of Cr supplementation during resistance training on body composition from previous studies [34,35,41,42]. Rather, the finding may be due to differences in the training regimens used. The heavy resistance-training programme of these previous studies aimed to improve body composition; therefore, the program utilized all major muscular groups. Moreover, Macdonald et al. [11] indicated body mass increased after 3 weeks of resistance training but not with complex training.

After the 4-week complex training, a significant improvement was observed in 30-m sprint performance in both the Cr and Pla groups, and no difference was noted between groups. The height and peak power of CMJ exhibited a significant time main effect; therefore, the complex training programme improved the performance of 30-m sprint and CMJ. Studies showed that the combination of Cr supplementation and resistance training increased the superior sprinting and jumping performance [43,44]. The inconsistent result may be attributed to different training programmes. The effects of complex training on producing superior chronic adaptations for explosive sports performance compared with other training methods have been summarized [9,32,33]. Juárez et al. [9] indicated that complex training exhibits increased effectiveness in spring and CMJ compared with conventional training (strength and power work in separated time). This finding may be attributed to the benefits of PAP on eliciting better transfer of the strength gain to running and jumping performances. Moreover, an appropriate individual rest interval may optimize the effectiveness of complex training.

The finding of our study supported the hypothesis that Cr supplementation during complex training would reduce muscular damage. In the current study, muscle damage was induced at the first and the last training bouts in both groups. However, the loading phase of creatine supplementation (20 g per day for 6 days) before the complex training significantly reduced CK activities 24 and 48 h after the complex training bout. The small effect size at the 48-hour time point may due to CK rising after exercise are highly variable and affected by individual factors [17]. Moreover, this effect can be sustained to the last training bout of 4 weeks during the training period when the maintenance phase of creatine (2 g per day) was consumed. The result of this study is consistent with previously studies that reported Cr supplementation exerted beneficial effects on reducing the biomarkers of muscle damage following repeated bouts of damaging exercise [20,21]. However, the effects of Cr supplementation on skeletal muscle damage and recovery from stressful exercise are discrepant. Some studies reported that Cr supplementation did not attenuate muscle damage on a maximal eccentric contraction protocol (50 and 150 reps) [23,24] or a high-repetition squat test (5 sets of 15 to 20 reps at 50% 1-RM) [22]. Repeated bouts of exercise result in protective adaptation given the repeated bout effect (e.g., neural, mechanical and cellular adaptations) in attenuation of muscle damage [45]. Thus, the reason for the inconsistencies may be related to the difference exercise protocols used. The mechanisms by which Cr supplementation decreased the biomarkers of muscle damage may be due to a combination of the multifaceted functions of Cr, including binding to phospholipid heads, stabilizing the membrane phospholipid bilayer, decreasing membrane fluidity and producing a more ordered state in the membrane [21]. Therefore, the study of Rahimi et al. [46] indicated that Cr supplementation reduced oxidative DNA damage and lipid peroxidation induced by a single bout of resistance training (7 sets of 4 exercise using 60–90 1-RM for gaining maximal strength) and recommended that coaches or athletes should consider using Cr supplementation in resistance-training protocols.

## 5. Conclusions

This study suggests that 4 weeks of complex training with optimal individual post-activation potentiation time (3 time per week) during the off-season period could reduce body fat percentage and enhance maximal muscular strength, performance of 30-m sprint, jump height and peak power. Moreover, creatine supplementation during the complex training protocol applied in this study could increase maximal muscular strength after 4 weeks of training and reduce the muscle damage caused by the complex training bout. Conditioning coaches may apply the results of this study to provide a proper training strategy to enhance performance.

## Figures and Tables

**Figure 1 nutrients-10-01640-f001:**
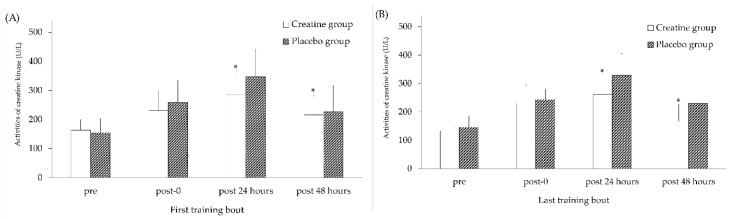
Creatine kinase activity in the creatine and placebo groups after (**A**) the first and (**B**) last training bouts. All values after training were significantly increased compared with pre-training values in both groups (*p* < 0.05). * Indicates a significant difference (*p* < 0.05) from the placebo group at the same time point.

**Table 1 nutrients-10-01640-t001:** Subject characteristics.

Variable	Creatine Group	Placebo Group
Height (cm)	171.93 ± 4.86	175.93 ± 8.49
Weight (kg)	67.86 ± 6.72	70.21 ± 11.16
Age (years)	20 ± 2	20 ± 1

Data are mean ± standard deviation (SD), *n* = 15 in each group.

**Table 2 nutrients-10-01640-t002:** Complex training protocol.

Set	Heavy Resistance Exercise		Recovery Time	Plyometric Training	
	Mode	Load	Rest Time	Mode	Repetitions
1	Half squat	5-RM	Optimal individual PAP time	Vertical jump	8
			4 min		
2	Half squat	5-RM	Optimal individual PAP time	Squat jump	8
			4 min		
3	Half squat	5-RM	Optimal individual PAP time	Vertical jump	8
			4 min		
4	Half squat	5-RM	Optimal individual PAP time	Squat jump	8
			4 min		
5	Half Squat	5-RM	Optimal individual PAP time	Vertical jump	8
			4 min		
6	Half squat	5-RM	Optimal individual PAP time	Squat jump	8

RM = repetition maximum; PAP = post-activation potentiation.

**Table 3 nutrients-10-01640-t003:** Body composition and sport performances in the creatine and placebo groups after 4 weeks of complex training.

Variable	Creatine Group		Placebo Group	
	Pre-Training	Post-Training	Pre-Training	Post-Training
Body mass (kg)	67.87 ± 6.72 (63.00–72.75)	68.51 ± 6.50 (63.79–73.24)	70.21 ± 11.16 (65.34–75.09)	70.34 ± 10.82 (65.62–75.06)
Body fat (%) ^a^	15.78 ± 4.18 (13.52–18.05)	13.77 ± 4.01 (11.87–15.68)	13.67 ± 4.37 (11.87–15.68)	12.76 ± 3.13 (10.86–14.67)
Fat-free mass (kg)	57.07 ± 4.84 (53.06–61.08)	58.97 ± 5.18 (55.03–62.91)	60.47 ± 9.55 (56.46–64.48)	61.58 ± 9.17 (57.64–65.52)
30 m sprint (second)	4.33 ± 0.20 (4.21–4.43)	4.14 ± 0.18 ^#^ (4.05–4.23)	4.27 ± 0.21 (4.16–4.38)	4.19 ± 0.16 ^#^ (4.10–4.28)
Half squat 1-RM (kg)	133.67 ± 14.07 (125.76–141.573)	178.33 ± 16.86 ^#,^* (169.99–186.68)	131.67 ± 15.77 (123.76–139.57)	165.66 ± 14.62 ^#^ (157.32–174.02)
Jump height (cm) ^a^	45.60 ± 5.18 (42.63–48.58)	54.60 ± 5.95 (51.63–57.59)	46.99 ± 6.04 (44.02–49.97)	53.93 ± 5.29 (50.96–56.91)
Jump peak power (W) ^a^	3729.40 ± 474.54 (3446.08–4012.72)	4166.52 ± 466.87 (3868.38−4464.66)	3903.41 ± 635.71 (3620.10–4186.73)	4260.44 ± 646.17 (3962.30–4558.58)

Data are mean ± SD (95% confidence interval (CI)). *n* = 15 in each group. ^#^ Indicates a significant difference (*p* < 0.05) from the pre-training value within the group. * Indicates a significant difference (*p* < 0.05) from the placebo group at the same training time. ^a^ Indicates a significant time main effect (*p* < 0.05).

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
