# Peer review of "Effects of 4-Week Creatine Supplementation Combined with Complex Training on Muscle Damage and Sport Performance"

_nutrients, 2018, doi:10.3390/nu10111640_

Reviewer 1 Report

Dear authors I read "Effects of 4-week Creatine Supplementation Combined with Complex Training on Muscle Damage and Sport Performance".

Is a current topic in sport nutrition. The manuscript is well written, but the reviewer have 1 question before pubblication:

- the athletes enrolled was in-season or in a off-season period during the study? If the athletes was in a off-season periodo please mention this aspect in discussion section

Author Response

General Comment:

The athletes enrolled was in-season or in an off-season period during the study? If the athletes was in an off-season period, please mention this aspect in discussion section.

Response: Thank you for your valuable suggestions. The athletes enrolled was in a off-season period during the study. We have reinforced the statements into the text. (the red color in lines 113-114, 232, 239, 303)

Reviewer 2 Report

This is an interesting and generally well-written manuscript. However, the introduction needs to provide a stronger rationale for the study, and in particular, the hypothesis of reduced muscle damage.

Abstract:

Support with actual data ad statistics

Line 15: A number of studies and review articles suggest that creatine supplementation enhance muscle strength and performance

Introduction:

Need to highlight the individual and varied nature of CK - https://www.ncbi.nlm.nih.gov/pubmed/23319463

Methods:

Line 90: Include participant characteristics for each group, rather than present in the results section. Present age as an integer.

Establishing the reliability of the measures would enhance the results as this would identify whether the changes are meaningful.

Include a sample size calculation. It would also be useful to calculate and present effect sizes and 95% confidence intervals.

Discussion:

There is plenty of useful information presented, although it could be more focused on the outcomes of the present study. Furthermore, the inclusion of some practical applications would be a beneficial addition.

Author Response

General Comment:

This is an interesting and generally well-written manuscript. However, the introduction needs to provide a stronger rationale for the study, and in particular, the hypothesis of reduced muscle damage.

Response: Thank you for your positive response and valuable suggestions. We have rewritten the sentences into the manuscript. (the red color in lines 80-84)

 Point-1:

Abstract: Support with actual data ad statistics. Line 15: A number of studies and review articles suggest that creatine supplementation enhance muscle strength and performance.

Response: Thanks you for kind reminder. Studies demonstrate that chronic Cr supplementation in combination with “resistance training” enhance muscle strength and performance (lines 40-42). The main purpose of this study is to investigate another training method, complex training, which involves the execution of heavy resistance exercise followed by a biomechanically similar explosive exercises are executed (lines 43-46). Therefore, the authors attempted to emphasize the studies of creatine supplementation and “complex training” in the sentences. In other words, we want to express that previously studies had investigated the effect of creatine supplementation on “an acute complex training bout”, but not on “long term complex training”. To avoid the misunderstanding, we rewrite the sentences to be “Creatine supplementation has an ergogenic effect in an acute complex training bout, but the benefits of chronic creatine supplementation during long-term complex training remain unknown.” (the red color in lines 16-17)

 Point-2:

Introduction: Need to highlight the individual and varied nature of CK - https://www.ncbi.nlm.nih.gov/pubmed/23319463.

Response: We added more description about the creatine kinase into the introduction. (the red color in lines 73-78)

 Point-3:

Line 90: Include participant characteristics for each group, rather than present in the results section. Present age as an integer.

Response: We cancelled the section of “3.1. Subject characteristics” and moved the “subject characteristics for each group to the section of “2.2. Subjects” and Table 1. Moreover, the age had presented as an integer. (the red color in lines 112 and 120)

 Point-4:

Establishing the reliability of the measures would enhance the results as this would identify whether the changes are meaningful.

Response: We have referred to the studies of Loturco et al. (2015) and Enoksen et al. (2009) to support the reliability of the jump performance and sprint tests. (the red color in lines 139-140, 151-152)

 Point-5:

Include a sample size calculation. It would also be useful to calculate and present effect sizes and 95% confidence intervals.

Response: We have added the statements of sample size calculation, effect sizes and 95% confidence intervals into the manuscript. (the red color in lines 108-111, 182, 189-191, 197-202, 208, 219-222)

 Point-6:

Discussion: There is plenty of useful information presented, although it could be more focused on the outcomes of the present study. Furthermore, the inclusion of some practical applications would be a beneficial addition.

Response: Thank you for your valuable suggestions. We describe the training season clearly. The athletes enrolled was in a off-season period during the study. We have reinforced the statements in the text to focus the outcome and clearly the practical applications (the red color in lines 113-114, 232, 239, 281-282, 303).
